# Deciphering the Molecular Interplay Between RXLR-Encoded *Avr* Genes and NLRs During *Phytophthora infestans* Infection in Potato: A Comprehensive Review

**DOI:** 10.3390/ijms26178153

**Published:** 2025-08-22

**Authors:** Bicko S. Juma, Olga A. Oxholm, Isaac K. Abuley, Chris K. Sørensen, Kim H. Hebelstrup

**Affiliations:** Department of Agroecology, Flakkebjerg Research Centre, 4200 Slagelse, Denmark; bickosnr@agro.au.dk (B.S.J.); olgaa@agro.au.dk (O.A.O.); ikabuley@agro.au.dk (I.K.A.); chris.sorensen@agro.au.dk (C.K.S.)

**Keywords:** *Phytophthora infestans*, RXLR effector, NLR, late blight, *Rpi* gene, gene stacking, *PiAvr*

## Abstract

Potato (*Solanum tuberosum* L.) is a globally significant staple crop that faces constant threats from *Phytophthora infestans*, the causative agent of late blight (LB). The battle between *Phytophthora infestans* and its host is driven by the molecular interplay of RXLR-encoded avirulence (*PiAvr*) effectors and nucleotide-binding leucine-rich repeat (NLR) immune receptors in potato. This review provides a comprehensive analysis of the structural characteristics, functional diversity, and evolutionary dynamics of RXLR effectors and the mechanisms by which NLR receptors recognize and respond to them. The study elaborates on both direct and indirect modes of effector recognition by NLRs, highlighting the gene-for-gene interactions that underlie resistance. Additionally, we discuss the molecular strategies employed by *P. infestans* to evade host immunity, including effector polymorphism, truncation, and transcriptional regulation. Advances in structural biology, functional genomics, and computational modeling have provided valuable insights into effector–receptor interactions, paving the way for innovative resistance breeding strategies. We also discuss the latest approaches to engineering durable resistance, including gene stacking, synthetic NLRs, and CRISPR-based modifications. Understanding these molecular mechanisms is critical for developing resistant potato cultivars and mitigating the devastating effects of LB. This review aims to bridge current knowledge gaps and guide future research efforts in plant immunity and disease management.

## 1. Introduction

Potato (*Solanum tuberosum* L.) is a crucial staple crop globally [1]. Originating from the highlands of the Andes in South America, where it was domesticated from wild tuber-bearing species, potato has evolved into a versatile crop cultivated across diverse climates and terrains worldwide [2]. Its adaptability and genetic diversity not only sustain millions of people but also serve as a vital source of income for small-scale farmers, particularly in low- and middle-income countries [3]. However, the cultivation of potato is constantly threatened by many pathogens and pests, among which *Phytophthora infestans* (Mont.) de Bary is a major challenge in many parts of the world [4]. *P. infestans* is a hemibiotrophic oomycete pathogen that causes potato late blight (LB), a disease that causes significant yield losses in potatoes. The symptoms of LB manifest as water-soaked lesions on foliage, rapid necrosis, and plant death (Figure 1) [5]. Particularly alarming is the pathogen’s ability to infect potatoes at any growth stage, even post-harvest (Figure 1D), posing a perpetual challenge to farmers worldwide [6]. The historical significance of *P. infestans* is evidenced by its catastrophic role in the Irish Potato Famine of the late 19th century, which led to widespread famine and mass migration [7].

The infection cycle of *Phytophthora infestans* begins with multinucleate sporangia that disperse via wind or rain splash [8]. These structures germinate directly or release uninucleate, motile zoospores that encyst on the host surface after shedding their flagella [9,10]. The cysts germinate and penetrate host tissues through stomata or via an appressorium-like structure. Once inside, a primary infection vesicle forms, leading to the development of intercellular hyphae and digit-like haustoria [11]. These haustoria preserve host plasma membrane integrity while enabling the delivery of RXLR effectors that subvert host cellular defenses. This transition from biotrophy to necrotrophy underpins the pathogen’s hemibiotrophic lifecycle [12,13].

To prevent the yield loss caused by late blight, farmers rely on the plants’ cell-autonomous immunity, as well as chemical fungicide application [14]. Potato and other wild relatives rely on two types of innate immune receptors to deploy efficient defense responses and impart disease resistance. At the extracellular level, cell-surface pattern recognition receptors (PRRs), including receptor-like kinases (RLKs) and receptor-like proteins (RLPs), perceive conserved microbial molecular structures called pathogen-associated molecular patterns (PAMPs) and elicit defense responses known as PAMP-triggered immunity (PTI) [15]. This leads to broad-range resistance towards non-adapted pathogens [16]. However, PTI is suppressed by effectors that the pathogens secrete into the host cells, leading to progression of the infection. In response, plants evolved intracellular receptors, resistance (*R*) genes, which are ubiquitous and most of which belong to the nucleotide-binding leucine-rich repeat receptors (NLRs). The NLRs either directly or indirectly detect effectors and initiate effector-triggered immunity (ETI). ETI signaling often results in a hypersensitive response (HR), a localized programmed cell death, that restricts pathogen proliferation [17]. The extensive use of chemical fungicides currently constitutes the primary control measures against LB in the fields. However, the reliance on fungicides for disease control presents environmental and health concerns. Moreover, the emergence of fungicide-resistant strains complicates control efforts [18].

Despite advancements in disease-management strategies, LB continues to have a significant toll on potato cultivation globally [19,20]. Economic losses due to fungicide costs, decreased yields, premature harvest, post-harvest secondary infection, and reduced tuber quality have been estimated to be around 9 billion euros globally [16]. Therefore, there is a need for the development of varieties that are resistant to *P. infestans*. In response, extensive studies have been conducted to elucidate the genetic and molecular mechanisms governing the interaction between *P. infestans* and its host plant [21,22].

This interdisciplinary research yielded significant results, particularly in identifying dominant resistance genes against *P. infestans* (*Rpi* genes) in wild potato relatives. In an early study, late-blight-resistance genes from *Solanum demissum*, a wild Petota *Solanum* species, were employed. Among them were the race-specific genes *R1*, *R2*, *R3a,* and *R3b*, which were introduced into cultivated potato and played a role in identifying the gene-for-gene relationship with *P. infestans* races [23]. The Rpi genes were introduced into commercial potato cultivars using traditional breeding procedures, but unfortunately, they were swiftly overcome by new strains of pathogens [24]. Since then, geneticists have identified more than 70 Rpi genes from wild potato relatives. The introgression of *Rpi* genes into commercial cultivars using traditional means such as crossing was faced with the difficulty of being time- and resource-intensive, especially for species separated from potatoes by crossing barriers such as varying endosperm balance numbers (EBNs) [25]. One noteworthy instance was the introduction of *Rpi-blb2*—which took more than 45 years—from *S. bulbocastanum* to the potato cultivars Bionica (2005) and Toluca (2004) [25]. However, using advanced genetic tools, which have significantly shortened introgression time, some of the *Rpi* genes were also cloned in potato cultivars to confer resistance to a high diversity of LB genotypes. An example is the *RB/Rpi-blb1* gene cloned from *S. bulbocastanum*, which has been transferred into *S. tuberosum* cv. Impala [26]. A thorough grasp of the *R/Avr* relationship helps guide successful LB resistance breeding and engineering.

In this review, we provide a comprehensive overview of the molecular interplay between RXLR effectors and NLRs during *P. infestans* infection in potato. Through an in-depth analysis of pathogen RXLR effectors, as well as the plant NLR immune receptors, this review aims to unravel the intricacies of the plant–pathogen interactions underlying LB resistance. In doing so, we hope to contribute to a better understanding of the genetic basis of LB resistance in potatoes for the development of new tools and methods to fight LB.

## 2. RXLR Effectors: Key Players in Pathogen Virulence

### 2.1. Structural Characteristics of RXLR Effectors

During the co-evolutionary arms race between plants and pathogens, pathogens evolved sophisticated arsenals to manipulate host cells during plant–pathogen coevolution. Oomycete pathogens employ secreted RXLR effectors, named for their conserved N-terminal Arg-X-Leu-Arg (RXLR) motif [27]. These small-secreted proteins, typically spanning 100 to 200 amino acids, play a crucial role in the virulence strategy of pathogens like *P. infestans* [28]. The pathogen’s genome contains hundreds of effectors, some of which may contribute to pathogenicity in a redundant manner [29,30]. These effectors are encoded by 563 genes belonging to the RXLR and 196 genes belonging to the crinkler (CRN) family [27]. However, to date all experimentally verified *P. infestans* avirulence genes encode RXLR effectors [31]. *P. infestans*’ repertoire of RXLR effectors is extremely varied and constantly changing. The immune system of the host plant exerts selective pressure, which drives variety and creates an ongoing arms race between the pathogen and the host [32]. The N-terminal RXLR motif functions as a molecular tag, facilitating translocation of the RXLR effectors into the cytoplasm of the host cell, a vital stage in the establishment of infection [27]. The translocation mechanism of RXLR effectors has been a subject of extensive study. For instance, it has been shown that RXLR motifs mediate entry into the cell by interacting with phospholipids, particularly phosphatidylinositol-3-phosphate (PI3P), which is located on the inner leaflet of the host cell membrane [33]. In addition, the effector can also manipulate the host cell processes by targeting the host vesicle-trafficking machinery, thereby facilitating entry and accumulation inside the cell [34]. For instance, the *PexRD12/31* effector family associates with VAMP72 R-SNARE proteins and accumulate at the host–pathogen interface [35]. Other RXLR effectors like *Pi22926* suppress the host immune responses triggered by pathogen recognition. They dampen the immunity creating a more permissive environment for their entry and accumulation inside the host cell [32]. This interaction is crucial for the uptake of the effectors into the host cytoplasm. Furthermore, many RXLR effectors harbor diverse effector domains, which confer specific functional properties essential for their virulence [36].

Recent advancements in structural biology have unraveled the intricate architecture of RXLR effectors, revealing a modular organization that underscores their multifaceted roles in host–pathogen interactions. These modular proteins often comprise distinct domains, an N-terminal signal peptide, an N-terminal half, and a C-terminal domain, each dedicated to executing specific tasks during infection, including functioning in secretion, translocation into the host cells, and effector activities, respectively (Figure 2A) [27]. For instance, some domains may possess enzymatic activities, enabling RXLR effectors to modify host proteins or metabolites to the pathogen’s advantage [37]. Other domains may facilitate protein–protein interactions, allowing RXLR effectors to hijack host cellular machinery for their benefit [11]. One example of such structural diversity is the presence of the conserved EER (Glu-Glu-Arg) motif within RXLR effectors, which is indispensable for their translocation and virulence functions [27]. This motif acts as a molecular switch, orchestrating the delivery of RXLR effectors into the host cell cytoplasm and initiating a cascade of molecular events conducive to infection establishment. For instance, Bos, Armstrong [38], elucidated, through X-ray crystallography, the *PiAvr3a* effector structure, revealing a compact globular fold with a central alpha-helix surrounded by beta-strands. The effector domain inhibits the host protein CMPG1, hence restricting programmed cell death. Since it enables the pathogen to elude the host’s immune system, this contact is essential to the effector’s virulence function. According to computational modeling and structural predictions, *PiAvrblb1* (also known as *ipi01*) and *PiAvrblb2* (*ipi02*) adopt similar globular folds with conserved surface residues that are crucial for their interaction with the host resistance proteins *Rpi-blb1* and *Rpi-blb2*, respectively (Figure 2B,C) [39]. *PiAvrblb1* has three main domains, the RxLR-dEER motif, which acts to carry the protein into the host cell cytoplasm through binding the cell surface phosphatidylinositol-3-phosphate, the RGD RLK-binding motif, which is required for binding to host legume-type lectin receptor kinases and disruption of the attachments between the host plasma membrane and cell wall, and the conserved W motif, which is essential for triggering *RGA2/Rpi-blb1*-mediated cell death (Figure 2B) [11,40,41,42,43].

Win, Krasileva [45] showed that RXLR effectors from different pathogenic oomycetes species share a conserved structural fold with conserved sequence motifs (W, Y, and L) in their C-terminus that often form tandem repeats. A study by Bos, Armstrong [38], inferred the structure of *Pi04314* from homology modeling to show a modular architecture with distinct functional domains. The domains are involved in interacting with host cellular machinery to suppress immune responses and promote pathogen colonization.

### 2.2. Diversity and Evolution of RXLR Effectors in P. infestans

The genome of the potato LB pathogen *P. infestans* was sequenced in 2009 and is among the largest compared to other related oomycete pathogens, with a size of 240 Mb and 17,797 protein-encoding genes [27]. This large genome size is attributed to a substantial repertoire of repetitive elements, including transposable elements (TEs), which constitute approximately 74% of the genome. These elements are believed to contribute to genome plasticity, enabling the pathogen to rapidly adapt to avoid host recognition [46]. RXLR effector genes are dispersed throughout the genome, often located in repeat-rich, gene-sparse regions. This localization likely facilitates their rapid diversification and evolution [27]. The RXLR motif, is followed by a variable C-terminal domain that defines the effector’s specific function within the host [11]. Comparative genomic and transcriptomic studies have revealed significant variation in the RXLR effector complement among different *P. infestans* isolates, reflecting ongoing evolutionary pressures. For example, Cooke, Schepers [47] used RNA sequencing to analyze the *P. infestans* transcriptome during infection, finding that RXLR effectors are highly expressed during early infection stages, highlighting their role in establishing the pathogen within the host.

The evolutionary plasticity and adaptation of RXLR genes arises from various genetic mechanisms, including gene duplication, point mutations, insertions and deletions, and recombination events [9]. These processes generate numerous effector variants, each potentially capable of targeting different host proteins or circumventing distinct host defenses [48]. Gene duplication is a major driver of RXLR effector diversity, resulting in multiple copies of effector genes that can evolve new functions or regulatory patterns. Many RXLR effector genes in *P. infestans* belong to large multigene families, some comprising dozens of members. This redundancy allows the pathogen to experiment with new variations while preserving essential functionalities [49]. TEs also significantly contribute to diversity. These mobile genetic elements can insert themselves into or near effector genes, causing mutations or altering gene expression. Transposable element insertions have been shown to create new splice variants of RXLR effector genes, enhancing the functional diversity of these proteins [9]. Recent studies have highlighted specific instances of RXLR effector diversity. For instance, *PiAvr2*, a novel RXLR effector in *P. infestans*, which targets a specific receptor-like kinase in potatoes to suppress host immune responses, was identified. Significant sequence variation in this effector across different *P. infestans* isolates indicates ongoing adaptation to varying host genotypes.

Other studies suggest that RXLR effectors evolved through a combination of vertical inheritance and horizontal gene transfer (HGT), enabling the pathogen to infect a wide range of host species [50]. The presence of conserved motifs within the RXLR domain across various oomycete species indicates a shared evolutionary origin, while sequence variability outside the RXLR motif reflects adaptation to specific host targets [51]. In the case of *P. infestans*, HGT events have resulted in the exchange of RXLR effector genes [52]. These events can introduce new effector functions into the pathogen’s genome, enhancing its ability to infect new host species or overcome host resistance [53]. For instance, Wang, McLellan [54] identified *PiAvr2*, a novel RXLR effector in *P. infestans*, which targets a specific receptor-like kinase in potatoes to suppress host immune responses. Significant sequence variation in this effector across different *P. infestans* isolates indicates ongoing adaptation to varying host genotypes. Many effectors show signs of positive selection, particularly in regions that interact with host targets [13]. This evolutionary pressure results in a “molecular arms race” between the pathogen and its host, with each continually developing new strategies to outcompete the other [51]. To avoid detection, *P. infestans* must continually evolve new effector variants, just as plants evolve new immune receptors to recognize and respond to RXLR effectors. This ongoing co-evolution leads to the fast turnover of effector genes, with some genes gaining or losing function over relatively short evolutionary periods [9,55]. Recent phylogenetic analyses have classified the effectors into distinct subfamilies or clades based on sequence similarity, each with its own evolutionary history, suggesting functional diversification within this protein family [56]. For example, the effector *PiAvrblb2* interacts with the host protein BSL1 to suppress immune responses, illustrating how effectors evolve to target key host defense mechanisms [57]. *PiAvrblb1* and *PiAvrblb2* specifically interact with the potato resistance proteins *Rpi-blb1* and *Rpi-blb2*, respectively. These interactions trigger an HR in resistant potato varieties, providing effective defense against *P. infestans*. Armstrong, Whisson [58], reported the existence of multiple isoforms of *PiAvr3a*, each with different abilities to suppress potato immunity. The variations in the sequences of *PiAvr3a* allow it to evade recognition by different variants of the host R3a genes, enabling the pathogen to infect diverse potato genotypes. The specificity of these interactions highlights the co-evolutionary dynamics between pathogen effectors and host resistance genes [39]. The *PiAvr3a* effector rapidly evolves such new isoforms through point mutations and recombination [38,59].

### 2.3. Functional Categorization and Host-Targeting Specificity

The RXLR effector suppresses host immune responses by manipulating host signaling pathways, altering host metabolism, and modifying host cell morphology to create a favorable environment for pathogen colonization. These functions are mediated by specific host-targeting domains within the effectors [3].

Functional categorization of RXLR effectors has revealed distinct groups based on their mode of action and effector domain composition [35]. Some effectors act as virulence factors directly contributing to disease development, while others function in effector-triggered immunity (ETI) as avirulence (*PiAvr*) factors recognized by host resistance proteins [60]. Numerous RXLR effectors work by inhibiting the host’s immune system, which permits *P. infestans* to spread and prolong infection [61,62]. For instance, PAMP-induced cell death and immunological responses are suppressed by the RXLR effector *PiAvr3a*, which binds and stabilizes the host ubiquitin E3 ligase CMPG1 [63]. The prevailing and extended presence of E3 ligase CMPG1 leads to the inhibition of infestin-4 triggered programmed cell death, a key defense mechanism in plants, thereby enhancing pathogen survival and colonization [38]. *Pi17316* (*PiAvr1*) has been reported to target the MAP3K VIK cascade in potato, a critical component of the plant immune’s response. By interfering with MAP3K signaling, the effector suppresses the activation of defense genes, allowing the pathogen to establish infection [54]. To aid in pathogen colonization, some RXLR effectors alter the structure of the host cell. These effectors often target components of the cell wall or cytoskeleton to create an environment more conducive to pathogen proliferation. For instance, the effector *IPI-O1* disrupts the host cell-wall-associated kinase PRK1, compromising the integrity of the host cell wall and promoting pathogen ingress and colonization [40]. *PITG_23006* from *P. infestans* translocates into the host nucleus, where it interacts with transcriptional regulators to suppress the expression of defense-related genes. This nuclear localization and interaction are crucial for its ability to modulate host gene expression and promote pathogen virulence [64]. Another example is the effector CRN8, which targets and alters the host cytoskeleton. It interacts with microtubules and actin filaments modifying the cytoskeletal architecture, aiding in the intracellular movement and proliferation of the pathogen [65]. Other RXLR effectors modify host metabolic processes and signaling. The effector *PiAvrblb2*, for example, targets the potato protein BSL1, a phosphatase implicated in brassinosteroid signaling, causing altered hormone signaling and increased vulnerability [66]. Another effector, *Pi04089*, binds and inhibits the host protease RCR3, which is involved in the activation of immune responses, inhibiting the activation of immunological responses, allowing the pathogen to evade detection and establish infection [67]. Another example is the effector *PiAvr2*, which interacts with the potato immune receptor *R2*, triggering a defense response that can be circumvented by certain effector variants, highlighting the dynamic co-evolution between pathogen effectors and host immune receptors [54,58].

## 3. NLR Proteins: The Guardians of Plant Immunity

### 3.1. Structural Organization and Domain Architecture of NLR Proteins

Most of the plant resistance (*R*) genes are members of a large gene family that encodes nucleotide-binding leucine-rich repeat (NB-LRR). Most potato species are tetraploids with 48 chromosomes (2n) and a DNA length of approximately 840 Mb (haploid), and 1–3% of the genes in the genome are *R*-genes, which are divided into two classes [68] (reviewed in Paluchowska, Sliwka [6]). One class of these *R* genes is the intracellular receptors that recognize pathogen effectors that have been delivered into the plant cell NLR. As multidomain proteins, they typically harbor an N-terminal domain, a highly conserved central nucleotide-binding (NB) domain, and a C-terminal leucine-rich repeat (LRR) domain. The other class are the receptor-like kinases (RLKs), which are considered extracellular receptors; however, their structure contains the extracellular domain, which typically consists of leucin-rich repeats or other motif-rich regions that interact with specific pathogen molecules, the transmembrane domain that anchors the protein in the cell membrane, and the intracellular kinase domain that initiates downstream signaling once the pathogen is recognized [69].

The N-terminal domain is involved in the regulation of signal transduction pathways for downstream defense responses, following recognition of the pathogen [70]. The domain varies among different NLR proteins and is classified into four groups: Toll-interleukin-1 receptor domain (TIR), or coiled-coil (CC) domains, with two subsets of the last ones; resistance to powdery mildew 8 (RPW8); and the G10-type CC-NLRs (CC_G10_-NLRs) (Figure 3) [71,72,73]. This domain often serves as a sensor that is responsible for recognizing pathogen-derived effectors or other signals, which are indicative of pathogen presence. The TIR domain is mostly involved in mediating the protein–protein interactions necessary for signal transduction, while the CC domain facilitates the oligomerization of NLR proteins and is critical for their activation and function [26]. An analysis of deletion in the CC or TIR domains has shown that these domains are required for signaling and subsequent cell death responses [74,75].

The NB domain is involved in nucleotide and ATP or GTP binding and hydrolysis, crucial for the activation of NLR proteins. In the absence of the pathogen effector, intramolecular binding domains or extramolecular binding with additional proteins will result in a closed and inactive conformation. The domain consists of three subdomains, NB, ARC1, and ARC2 (reviewed by [76]). The NB domain acts as molecular switch, regulating the activation state of the NLR protein by inducing a conformational change via nucleoside triphosphate (NTP, such as ATP and GTP) hydrolysis to regulate signal transduction. In addition, it consists of eight highly conserved motifs, including RNBS-A and RNBS-B, P-loop, kinase-2, and GLPL motifs. These motifs have different functions. The P-loop motif binds ATP/ADP and contributes to the nucleotide-dependent conformational changes of NLR proteins. Kinase-2 and GLPL motifs are involved in nucleotide binding and hydrolysis, stabilizing the ARC domain in its active or inactive form (Figure 4A) [76]. A study by Foster, Park [77], on the potato NLR protein Rpi-vnt1 reported that mutations in the NB-ARC domain disrupt its ability to bind nucleotides, thereby impairing its functionality in the resistance response [78].

The LRR domain, located at the C-terminus, which contains a series of leucine-rich repeats that form a solenoid structure, is involved in protein–protein interactions and ligand recognition. It is responsible for effector recognition and the subsequent activation of immune responses. The LRR domain comprises a 2–42 tandem repeat of amino acids, which adopt a β-sheet structure and adjacent loop regions, that is required for structural arrangement of the domain. The domain is considered the main determinant in pathogen recognition specificity and functions directly or indirectly in the binding of *PiAvr* gene products. While it has been hypothesized that the ARC2 subdomain relays pathogen recognition at the LRR domain into conformational changes that result in downstream signaling, the LRR domain generally appears to interact with the ARC1 subdomain. The activation upon interaction with the CC-NB-ARC portion is thought to require the full LRR domain [79]. The removal of this domain has been shown to result in loss of effector recognition, as well as an increase in autonecrosis in NLRs, indicating an additional role for the LRR domain in NLR autoinhibition [75,79]. For instance, Monino-Lopez, Nijenhuis [80], reported that CRISPR/Cas9-induced frameshifts in the LRR domain of candidate B2-3 (*Rpi-chc1.1*) lead to *Phytophthora* susceptibility.

With the advent of resistance gene enrichment sequencing, 755 full- and partial-length NLR proteins have been identified from the sequences of the *Solanum tuberosum* Group Phureja clone DM1-3516 R44 reference genome (Table 1) [81]. To date, more than 50 *R*-genes conferring resistance specifically to *P. infestans* (*Rpi* genes) have been cloned from *Solanum* species, with the majority belonging to the CC-NLR family, but several still remain unclassified [71]. The longest *Rpi* gene includes *Rpi-amr1-2307* (7277 bp) from *S. americanum* and *Rpi-blb2* from *S. bulbocastanum* (4858 bp) belonging to the CC-NLR class, and *R1* from *S. demissum* (4102 bp), which is part of the LZ-NB-LRR group. On the other hand, the shortest genes are *R2* family members, such as *R2* from *S. demissum* (2538 bp), *Rpi-edn1.1* from *S. edinense* (2544 bp), *Rpi-hjt1.1*, *Rpi-hjt1.2*, and *Rpi-hjt1.3* from *S. hjertingii* (2544 bp) [6].

### 3.2. Mechanisms of NLR Activation and Signaling in Plant Immunity

Structural studies have provided insights into the mechanisms of NLR protein activation and signaling. In the inactive state, NLR proteins are maintained in a closed conformation through intramolecular interactions, particularly involving the LRR domain. Upon effector recognition, conformational changes occur, leading to the exposure of the NB domain and activation of downstream signaling pathways (Figure 4) [82].

NLR proteins are activated in response to pathogen-derived effectors, delivered into plant cells during infection, and the activation mechanism involves a series of conformational changes [83]. It is anticipated that there would be notable differences between the molecular mechanisms that activate sensor NLRs and those of their helpers, which are yet poorly known [84]. Nonetheless, multimerization via the NB-ARC domain after the exchange of ADP for ATP appears to be a crucial stage in NLR activation, as suggested by their similar evolutionary origins and shared domains. NLRs are frequently referred to as molecular switches in immune signaling because of this. In the ADP-bound state, thought of as the “OFF STATE”, the NLR is stabilized in the inactive state by the LRR’s association with the NB-ARC domain, preventing NLR autoactivation [85]. NLR activation, often known as the “ON STATE”, is typically connected to the ATP-bound state. Ligand binding of the LRR releases the NB domain, allowing ADP-ATP exchange via the ATPase activity of the NB-ARC domain itself or through NLR-interacting proteins, acting as nucleotide-exchange factors, and ATP binding promotes the oligomerization of NLRs (Figure 4A) [86]. With the advent of new technologies, biophysics and cryo-electron microscopy, it has been shown that the activation of NLR proteins leads to the formation of a “resistosome” complex, composed of an activated NLR protein, a co-receptor, and an effector protein [87,88]. The structure of the resistosome was first revealed through characterization of HOPZ-ACTIVATED RESISTANCE 1 (ZAR1), an ancient CC-NLR, which emerged in the Jurassic period, conserved across flowering plant species. This complex initiates downstream signaling events, including the activation of defense responses such as callose deposition, ROS production, and defense gene upregulation [85]. Oligomerized NLRs subsequently initiate downstream signaling events, mediated by the N-terminal signaling domain.

## 4. Recognition Mechanisms Between RXLR Effectors and NLRs

### 4.1. Direct and Indirect Modes of Effector Recognition by NLRs

According to the gene-for-gene model of immunity, NLR proteins can either recognize pathogen effectors through direct physical interaction (protein–protein interaction) or indirectly through the ‘guard hypothesis’, which means recognizing a modification in the host protein targeted by the effectors; moreover, the indirect mode has two subcategories, guard model and decoy model (Figure 4B,C) [89,90]. Starting with the direct mode, NLRs interact with pathogen effectors as a singleton, leading to their activation and the initiation of downstream immune responses (Figure 4A). This interaction often occurs through specific protein–protein interactions mediated by domains within the NLR protein, such as the LRR domain [91]. A yeast-two hybrid assay of several NLR–effector combinations experimentally prove the direct NLR–effector interactions as the underlying resistance specificity in some diseases. The binding triggers a conformational change in the NLR, leading to its activation. The activating oligomerization of the receptor proteins leads to the subsequent initiation of a downstream signaling pathway, including the activation of mitogen-activated protein kinases (MAPKs) and the production of reactive oxygen species (ROS), ultimately leading to a hypersensitive response (HR) and localized cell death to contain the pathogen [54]. In the case of the *P. infestans*–NLR interaction, the most common example with evidence of direct physical interaction-based recognition is the *IPI-O class 1* family of RXLR effectors, such as *PiAvrblb1*, which contains a conserved W motif in its C-terminal domain that is recognized by *Rpi-blb1* (*RB*) from *S. bulbocastanum*, triggering cell death (Figure 5) [39,92]. *IPI-O1* in hosts that do not contain *Rpi-blb1* binds to lectin receptor kinase (LRK), thereby disrupting cell wall–plasma membrane adhesion [43]. The binding between the *Rpi-blb1* receptor and *PiAvrblb1* occurs via the LRR region. The exact binding involves a precise molecular fit, allowing the NLR to detect the effector with high specificity. Upon binding, *Rpi-blb1* undergoes a conformational change where the NB domain switches from an inactive to an active state, promoting ATP binding and hydrolysis within the NB-ARC domain. Once in the active state, *Rpi-blb1* interacts with EDS1 and NDR1 pathways to amplify the immune response, which includes inducing ROS production, callose deposition at the infection site, and transcriptional reprogramming to activate defense-related genes leading to HR and further signaling molecules, such as salicylic acid, which helps in sustaining and amplifying the immune response across the plant (Figure 5). Another example of direct interaction of the *Rpi* gene with its cognate effector is *R3a* from *S. demissum*, which recognizes the *PiAvr3a^KI^* effectors [93]. *PiAvr3a* is a well-studied RXLR effector, where two major alleles exist: *Avr3a^KI^* (carries Lys80/Ile103 in mature protein), which triggers *R3a*-mediated HR, and *Avr3a^EM^* (Glu80/Met103), which is virulent allele. In the virulent state, *Avr3a^EM^* binds to the potato U-box E3 ubiquitin ligase CMPG1 and stabilizes it, thereby suppressing immune signaling and INF1-triggered cell death. Thus, *Avr3a* is essential for full pathogen virulence by dampening both PTI and ETI; *R3a* overcomes this by recognizing *Avr3a^KI^.* Another example is *R1*, which is one of the earliest NLRs to be identified, conferring resistance against *P. infestans*. It recognizes the *PiAvr1* effector, and the recognition occurs through direct binding of the LRR domain to *PiAvr1*. The interaction induces a switch of *R1* to an ATP-bound active state, exposing signaling domains essential for downstream signaling [94,95]. A recent study by Du, Chen [96], identified a novel *PiAvr1* variant (named A-L (Avr1-like)) in *P. infestans*, which at the protein level, shows 82% homology to *PiAvr1*, and it can still be recognized by *R1* proteins, indicating the adaptability of this immune recognition system. In addition, one of the earliest identified *Rpi* genes, *R8* (*Rpi-Smira2*) (an Sw-5 homolog from *S. demissum*), which has been shown to offer broad spectrum and durable resistance, also interacts with its cognate *PiAvr8* directly [97,98]. *R8* acts by stabilizing the potato deSUMOylating isopeptidase (StDeSI2), positively regulating resistance, thereby resulting in localized programmed cell death. In the absence of the R8 gene, *PiAvr8* interacts with and destabilizes StDeSI2 through the 26S proteosome inhibitor prematurely attenuating PTI [99,100]. From the study by Ahn, Lin [101], it was shown that upon effector recognition, both *Rpi-amr1* and *Rpi-amr3* derived from *Solanum americanum* activate the formation of a high-molecular-weight complex of NRC2, dependent on a functional ATP-binding motif of both NLRs. *Rpi-amr3* activates the oligomerization of NRC2 and NRC4 resistosomes, while *Rpi-amr1* induces the formation of only the NRC2 resistosome [84].

Indirect recognition, on the other hand, involves the detection of effector-mediated modifications or perturbations in host targets (Figure 4B,C), via phosphorylation, proteolytic cleavage, or conformational changes in plant functional proteins (guard model) or in nonfunctional plant proteins. In both cases, this indirect detection triggers NLR activation and initiates immune responses. According to the guard model (Figure 4B), the *Rpi-vnt1* gene from *S. venturii* confers resistance to *P. infestans* by indirectly recognizing the *PiAvr-vnt1* effector. *PiAvr-vnt1* targets and binds to a full-length nuclear-encoded chloroplast protein, glycerate 3-kinase (GLYK), involved in energy production, leading to the activation of *Rpi-vnt1*, which is also dependent on the presence of light. In the dark, Rpi-vnt1.1–mediated resistance is compromised because plants produce a shorter GLYK—lacking the intact chloroplast transit peptide—that is the binding part of *PiAvrvnt1*. The transition between full-length and shorter plant GLYK transcripts is controlled by a light-dependent alternative promoter selection mechanism [102] (Figure 5). *Rpi-blb2*, which is another CC-NLR type *R* gene identified from *S. bulbocastanum*, recognizes and indirectly interacts with the *PiAvrblb2* (*PITG20300*) effector via the interaction with the host protein BAK1. *PiAvrblb2* usually accumulates at the perihaustorial plasma membrane where it prevents secretion of the cysteine proteinase C14 to the apoplast [103]. The protein BAK1 is manipulated by *PiAvr2* to inhibit the secretion of the host papain-like cysteine protease (PLCP) C14, which in turn suppress immune signaling (as depicted in Figure 5). The detection of these manipulations by *Rpi-blb2* activates immune responses to counter the pathogen [104,105]. Naveed, Bibi [106] showed that physical interaction between *PiAvrblb2* and calmodulin is required for its recognition by *Rpi-blb2*, suggesting that calmodulin serves as a guardee, monitored by *Rpi-blb2*, to induce the resistance response. Another notable example of an indirect interaction between NLR and the corresponding *PiAvr* protein is the interaction between the *R2* gene and the multiallelic highly divergent *PiAvr2*. *PiAvr2* is localized to the host nucleus and cytoplasm, but mainly at the perihaustorial plasma membrane. It mediates the interaction between the phosphatase BRI1-suppressor 1-like family of proteins (BSL1, BSL2 and BSL3) from potato and oomycete infestin 1 (INF1). The association of *Avr2*, BSL1, and *R2* corresponds to a three-molecule interaction which is described by the guard model [107] (Figure 5). The effector modulates the BSL1 host protein that interacts with *R2*. The modification is detected by and activates *R2*, triggering a defense response that includes a conformational change in NB-ARC and the exchange of ADP for ATP, HR, and the activation of defense-related gene expression [66]. This suggests that BSL-family proteins act as guardees and contribute to the *PiAvr2*-mediated virulence of *P. infestans* [107].

Another indirect model of recognition is the plant decoy strategy, where the monitored host factor has no measurable resistance function but serves as a bait to trap pathogen effectors that target structurally related basal defense components, thereby triggering ETI. The decoy plays no other role in the plant defense, simply serving to copy a true defense component effector target and attract the pathogen effector (Figure 4C) [108]. The effector-induced modification of the decoy will then be perceived by an NLR protein, with subsequent activation of the plant immune response. However, while this strategy has been reported in other crops, it is yet to be extensively elucidated in *S. tuberosum*. In addition to the Rpi genes and their cognate RxLR *Avr* effectors described in this section, Table 2 also summarizes these interactions with additional Rpi genes that have been recently cloned in wild *Solanum* species.

### 4.2. Mechanisms of Effector Evasion and Host Counterstrategies

#### 4.2.1. Effector Polymorphism

To date, numerous mechanisms have been identified that allow effectors to evade recognition by their corresponding R proteins. The products of unrecognized alleles usually act as the virulence factors. One of the methods could be single-nucleotide polymorphisms (SNPs), insertions, deletions, and recombination events, which can all result in amino acid changes in the effector protein sequence. These single amino acid differences do not affect the functionality of the effectors but could block host NLR proteins from recognizing the effector. This rapid evolution enables the pathogen to escape the plant’s immune system, allowing for effective infection and colonization.

Several polymorphic forms of *PiAvr2* exist, each with unique amino acid sequences that influence identification by the potato NLR protein R2. A study by Gilroy, Breen [119] revealed a difference in 13 different amino acid residues between homologous classes of avirulent *PiAvr2* and virulent *PiAvr2*-like effectors, eight of which are in the C-terminal effector domain, which determines recognition by R2. Yang, Liu [120] explored on the role of intrinsic disorder in the development of pathogenicity in the RXLR *PiAvr2* effector and found out that it exhibited high nucleotide diversity generated by a point mutation, early termination, an altered start codon, a deletion/insertion, and intragenic recombination. The virulent *PiAvr2* had a higher disorder tendency in both N- and C-terminal regions. These polymorphisms represent the dynamic co-evolution of pathogen effectors and host immunological receptors.

One of the best studied oomycete effectors is *PiAvr3a*. It occurs in two forms, differing in only three amino acids created by a point mutation: two of these changes, at positions 80 and 103, are present in the mature proteins lacking the signal peptide [121]. One of the forms, *PiAvr3a^KI^*, is efficiently recognized by *R3a*, when at the same time, the second one, *PiAvr3a^EM^*, is not, leading to infection [58,63,122].

*Rpi-blb2* resistance has shown great potential for improving resistance to *P. infestans* in field trials. However, it was found that the *PiAvrblb2* effector that is recognized by this gene is highly diverse and encodes effector variants that can avoid detection. One of them is a virulent variant (*PITG20303*) that harbors an amino acid alteration from Ala/Ile/Val to Phe at the position 69 [25]. From the study by Oliva, Cano [123], it was revealed that *PiAvrblb2* has evolved from a single-copy gene in a putative ancestral species of *Phytophthora* and recently expanded in the *P. infestans* that infects *Solanaceous* hosts. As a result, at least four variants of *PiAvrblb2* could be found in the present-day *P. infestans* population, indicating the potential benefit for the pathogen to preserve duplicated and functionally different versions of the gene. In 2021, Du, Chen [96] reported a new virulent variant of *PiAvrblb2* (PITG20300) that also escapes detection by *Rpi-blb2*.

Other RXLR effectors escape the detection by their cognate *Rpi* genes due to the presence of C-terminal truncation. This genetic diversity may be generated by point mutations and premature stop codons. However, there are examples showing that a protein truncation needs to be specific to have any effect on the virulence. van Poppel, Guo [124] reported that deletion of the RXLR-dEER domain neither enhances nor suppresses the elicitor activity of the *PiAvr4* proteins. In this example, it was found that the full function of the effector requires the presence of a Y-motif together with either a W1 or W3 domain [111]. This was confirmed when Waheed, Wang [125] showed that a frameshift caused by single base-pair deletion (ΔT^96^) generates two premature stop codons, truncating almost the entire C-terminal of the effector protein starting from the 97 amino acids before the W1 domain leading to a virulent phenotype, with potato plants carrying the *R4* gene.

Apart from effector polymorphisms, changes in the expression pattern and regulation could also result in the evasion of detection by corresponding Rpi proteins. In the absence of apparent genetic mutations, effectors have developed plasticity for when and for how long they are expressed during infection, to avoid detection. For instance, changes in expression of the *PiAvrvnt1* effector lead to the inability to be detected by *Rpi-vnt1* plants [115]. Monino-Lopez, Nijenhuis [80] reported that expression of the *PiAvrchc1.2* effector is rapidly downregulated in the first hours after inoculation with *P. infestans* isolates, which explains why the presence of *Rpi-chc1.2* in potato plants does not provide resistance to late blight.

#### 4.2.2. Taking the Side of the Plants in the Fight

Potato plants have evolved a variety of defense mechanisms to combat the pathogen’s evasion methods through the diversification of *R* genes via recombination, gene conversion, duplication, and/or selection [81]. One such technique is the diversification of NLR genes, which allows for the detection of a wide range of effectors. While some of the Rpi genes have been found to be race-specific and have rapidly become ineffective because of the evolution of pathogen effectors, the following have been reported to provide intermediate and broad-spectrum resistance against *P. infestans*: *Ri-blb1*, *Rpi-blb2* and *Rpi-blb3* from *S. bulbocastanum*, *R8* from *S. demissum*, *Rpi-vnt1.1* from *S. venturii*, *Rpi-sto1* from *S. stoloniferum*, and *Rpi-chc1.1* from *S. chacoense*. However, they have not been widely introduced into potato cultivars due to the crossing barriers. Another strategy is to deploy numerous NLR proteins that can recognize the same effector, resulting in redundancy in immune detection. This multilayer defense system ensures that even if one NLR is bypassed by the virus, the others can detect and respond to the infection. Aguilera-Galvez, Chu [126] showed that the *Avr2* effector proteins are not only recognized by the *R2* protein from *S. demissum* but also by *Rpi-blb3*, *Rpi-mcq1* from *S. mochiquense*, *Rpi-hcb1.1*, and *Rpi-hcb1.2* from *S. huacabambense*.

#### 4.2.3. Engineering and Pyramiding of R-Genes

Engineering *R*-genes is a means to confer plant recognition of the virulent effector forms of *P. infestans*. An example is *PiAvr3a^EM^*, the virulent allele of *PiAvr3a*, that avoids the recognition of *R3a* [127,128]. This allele is widely present in *P. infestans* isolates around the world, suggesting that it is important for *PiAvr3a* pathogenesis. The identification of potato varieties with the capability of recognizing a broad spectrum of *PiAvr3a* alleles is therefore predicted to be vital for improving the resistance against late blight [127]. In line with this, two potato cultivars, Qingshu9 and Longshu7, with a single *R* gene that recognizes the virulent *PiAvr3a^EM^* triggering an HR response have been identified. As an alternative, *R*-genes can be engineering to recognize virulent effector alleles. For example, it has been shown that a variant of *R3a* that was unable to recognize *PiAvr3a* could be turned into a functional variant (*R3a**) through error-prone PCR and iterative rounds of DNA shuffling of the LRR domain [127].

Another method for increasing plant resistance is through gene pyramiding, where resistance genes are stacked within a cultivar. This is an advantage since the resistance achieved by the insert of a single *Rpi* gene is often broken within short time by the evolving pathogen. Breeders can generate potato types with broad spectrum and long-lasting resistance to *P. infestans* by stacking various NLR genes that identify distinct effectors or variations of the same effector. This offers resistance and delays the pathogen infection for the whole planting season (90 days) [25]. For example, the pyramiding of *Rpi-sto1*, *Rpi-blb3*, and *Rpi-vnt1* genes has been demonstrated to offer broad resistance against various strains of *P. infestans* [129]. The stacking of Rpi genes with broad spectrum resistance against *P. infestans* pathogen is comprehensively explained in the review articles by Paluchowska, Sliwka [6] and Berindean, Taoutaou [130].

#### 4.2.4. NLR Clusters

NLR genes are often organized in clusters. Such clusters are made up of several NLR genes that are situated next to each other on a chromosome, allowing coordinated expression, functional synergy in the immune response and adaptation in the face of evolving pathogens. These clusters can rapidly change, via conversion and recombination, producing new NLR variants with unique recognition specificities. NLR gene clustering offers a genetic diversity reservoir that can be used to create potato types that are resistant due to their robustness and malleability in adapting to recognized pathogens. An example is the genes *Rpi-blb1*, *Rpi-blb2*, and *Rpi-blb3* from *S. bulbocastanum*, which are all organized into different clusters with allelic variations in different accessions [131]. Furthermore, plant NLRs have undergone functional specialization to play one of two roles: sensing pathogen effectors (sensor NLRs) or coordinating immunological signaling (helper or executer NLRs). Sensor NLRs recognize effectors directly, whereas helper NLRs serve as signaling hubs for many sensor NLRs, converting effector recognition into an effective plant immune response [91]. Interestingly, these functionally specialized NLRs often form gene pairs or clusters in plant genomes and function together in immune activation and regulation [132]. In addition to NLR pairings, genetically scattered NLRs frequently collaborate and form complex immune receptor networks. In solanaceous plants, CC-NLR proteins known as NLR-REQUIRED FOR CELL DEATH (NRC) operate as helper NLRs (NRC-H) for many sensor NLRs (NRC-S) to mediate immune responses and confer disease resistance against diverse pathogens [133]. Recent studies have focused on the construction of comprehensive NLR gene inventories, referred to as pan-NLRomes, which encompass the full diversity within a species [134]. For instance, in *Solanum americanum*, 71% of its NLR genes are clustered and the rest are singletons, indicating a rich reservoir for breeding programs aimed at enhancing disease resistance in cultivated potato [84].

### 4.3. Technological Advances in Studying RXLR-NLR Interactions

#### 4.3.1. Genome Sequencing and Pan-NLRomes

The availability of high-quality reference genomes of potato and its wild relatives such as *Solanum americanum* has been a game-changer in the study of RXLR effector–NLR interactions [84,135,136]. These genomes have revealed the extensive genetic variation in potato species and have enabled the construction of pan-NLRomes, which encompass the full diversity of NLR genes within a species [137]. High-quality reference genomes for *S. americanum*, a wild relative of potato, have been generated from four different accessions, allowing for comprehensive studies of NLR gene clusters. The resequencing of 52 accessions has led to the definition of a pan-NLRome, which encompasses the diversity of immune receptor genes within this species. This genomic resource opens the opportunity to screen for variations in the recognition of numerous RXLR effectors, facilitating the identification of specific NLR genes that confer resistance to *P. infestans*. Apart from *S. americanum*, phased genome assembly and de novo assemblies of other wild relatives have also been constructed for pan-NLRomes [138,139,140,141].

#### 4.3.2. Effector-Omics and High-Throughput Screening

With more than 560 RXLR effectors predicted computationally, assigning the biological functions for the majority of them has not been achieved to date [112]. Effector-omics, a technique involving transforming and transiently expressing RXLR recombinant plasmids into plant leaves to determine the existence of potential resistance genes in host materials based on their ability to trigger immune responses, has become a powerful tool for studying RXLR effector–NLR interactions [142,143,144]. By screening hundreds of RXLR effectors from *P. infestans* based on diverse *Solanum* accessions, it is now possible to map the effector-triggered immunity landscape and identify one-on-one effector–receptor interactions. For instance, Oh, Young [112] screened 62 infection-ready *P. infestans* RXLR effector clones, corresponding to 32 genes and assigned activities to several of the genes. The study successfully identified two *PiAvr* gene families which induced hypersensitive cell death in the presence of the *S. bulbocastanum*-resistance genes *Rpi-blb1* and *Rpi-blb2*, as well as novel elicitors and suppressors of cell death. The method has also been utilized to screen for resistance genes in wild relatives of potato. Lin, Jia [84] screened 315 RXLR effector proteins against 52 *S. americanum* accessions. The study led to the discovery of three new *Rpi* genes (*Rpi-amr4*, *R02860*, and *R04373*).

High-throughput screening approaches, such as yeast two-hybrid (Y2H), and co-immunoprecipitation assays have proven to be useful for determining the interactions between RXLR effectors and host proteins [145]. The approach relies on the principle of reconstituting a functional transcription factor by binding its two separate domains (DNA-binding domain and activation domain) into proximity through protein–protein interactions [145]. These approaches enable the quick screening of vast libraries of effectors and host proteins, resulting in a comprehensive map of effector–host interactions. For example, Y2H tests have been utilized to uncover interactions between RXLR effectors and host proteins implicated in immunological signaling, suggesting new targets for increasing resistance. An example is a study by Zhang, Li [146], which showed that Pi05910, an RXLR effector that is highly conserved and expressed at the early stages of *P. infestans* infection, interacts with glycolate oxidase NbGOX4 and its orthologue protein StGOX4 in potato. Wang, Trusch [116] also showed that the use of Y2H in combination with other assays helps in understanding the contribution of BSL phosphatases to R2-and Rpi-mcq1-mediated HR [147,148].

Protein microarrays are another method for high-throughput screening. Protein microarrays are made up of immobilized host proteins that can be probed with pathogen effectors to determine binding relationships [149]. This method has shown several interactions between RXLR effectors and critical host immune system components, suggesting potential resistance-enhancing targets. For instance, Bos, Armstrong [38] utilized a parallel expression and screening that mirrors a microarray method to show that *P. infestans Avr3a* associates with and stabilizes the host E3 ligase CMPG1.

#### 4.3.3. Structural Biology

Methods in structural biology have been used to map the binding sites of effectors onto host proteins, their interaction with the host DNA, and their impact on host cellular functions [150,151]. In addition, this field provides a deeper understanding into the host defense mechanism by providing insights into the three-dimensional structures of plant immune receptors and effector–target interactions and sheds light on the conformational changes that occur in immune receptors and downstream signaling molecules upon pathogen recognition, elucidating the molecular basis of signal transduction [152,153,154,155]. Advances in structural biology, notably cryo-electron microscopy (cryo-EM), X-ray crystallography, nuclear magnetic resonance (NMR) spectroscopy, and computational methods such as AlphaFold2, have revealed information about the molecular interactions between RXLR effectors and NLR proteins [99,118,156,157,158,159].

Cryo-EM has emerged as an effective and powerful method for analyzing big protein complexes with near-atomic resolution allowing for visualization of the effector–protein complexes in 3D [160]. For example, cryo-EM investigations have revealed the structure of the *PiAvr3a*–*R3a* complex. A recent study by Madhuprakash, Toghani [159] utilized cryo-EM to determine the structure of the potato NLR protein NbNRC2 in its resting and activated states, revealing that an effector from the pathogen bridges the ENTH-domain protein TOL9a to NRC2 to suppress immunity.

X-ray crystallography has also played a role in understanding the structure of RXLR effectors and NLR proteins. The approach involves growing crystals of the target protein, which are then exposed to X-rays [161]. This method gives high-resolution structural information, which is critical for understanding the precise interactions between effectors and NLRs. X-ray crystallographic analyses have revealed the structures of numerous RXLR effectors, including *PiAvrblb2* and *PiAvr2*, in complex with their host targets. For instance, Boutemy, King [37] utilized this method to show the crystal structures of the effector domains from two oomycete RXLR proteins, *Phytophthora capsica AVR3a11* and *Phytophthora infestans* PexRD2. Despite sharing <20% sequence identity in their effector domains, they display a conserved core α-helical fold. This core fold has since been recognized as the WY-domain, named after a conserved Trp-Tyr motif in one of the helices. These investigations discovered critical residues and structural motifs involved in effector function and recognition. A notable study utilized X-ray crystallography to investigate the interaction between the *P. infestans* effector protein PexRD54 and potato ATG8. The researchers defined the crystal structure of PexRD54, which included multiple domains and an ATG8 family-interacting motif (AIM). They also solved the crystal structure of potato ATG8CL in complex with a peptide from PexRD54, providing insights into how this effector perturbs host-selective autophagy during infection [162]. Furthermore, the structure of the *P. infestans* effector *PiSFI3* complex with a potato U-box kinase (StUBK) has been determined via X-ray crystallography for structure-guided mutagenesis, revealing how *PiSFI3* suppresses plant immunity [143]. It is worth noting that X-ray crystallography excels with small-to-medium-sized proteins that can be purified and crystallized. Many *RxLR* effectors are 100–200 amino acids (10–20 kDa) in their mature form, which is amenable to crystallization (as long as flexible regions can be trimmed or managed). Crystallography has thus been effectively applied to solitary effector domains (e.g., *Avr3a11*, *PexRD2*, *ATR1*, and *Avr_Pii* from other pathogens) and to effector–peptide complexes (e.g., ATG8–AIM peptide).

Nuclear Magnetic Resonance (NMR) spectroscopy is another tool that has contributed significantly to understanding the *RxLR* effector structure and dynamics, especially for proteins too small or too flexible for easy crystallization. One of the earliest structural analyses of a *P. infestans* effector was performed via NMR: the solution structure of AVR3a (lacking its signal peptide, ~10 kDa) [163]. The *Avr3a* NMR study produced a 3D structure, in addition to examining the accessibility of the RxLR motif and nearby regions to proteolysis and modification [163]. It demonstrated that the RxLR and dEER motifs are in a flexible, solvent-exposed loop, rationalizing how the effector-processing protease can cleave after the RxLR. This was an important confirmation of a biochemical model, showing NMR’s strength in linking structure to function in terms of protein processing.

Another group of tools are AI-driven protein structure prediction algorithms, such as AlphaFold2 (AF2) and ColabFold, which are revolutionizing the prediction of 3D protein structures with remarkable accuracy [164]. These tools have emerged as vital resources in resistance breeding by facilitating the study of plant–pathogen interactions at the molecular level, enabling the design of more effective resistance (R) proteins, and guiding the development of durable disease-resistant crops. AlphaFold2, developed by DeepMind, has revolutionized structural biology by predicting protein structures with remarkable accuracy using deep learning algorithms. This too has particularly been useful for predicting the structures of proteins that are difficult to study experimentally. ColabFold, on the other hand, is a streamlined, open-source version of AlphaFold2 that offers users faster and more accessible protein-structure-prediction capabilities. It is hosted on Google Colab and allows anyone to acquire high-quality structural predictions without the need for large computational resources [165]. ColabFold accelerates the discovery of novel R proteins, as well as the study of effector proteins from *P. infestans*, allowing for high-throughput predictions that may be used immediately for breeding programs. In the context of RXLR-NLR interactions, AlphaFold2 and ColabFold have been used to predict the structures of several NLR proteins and their potential complexes with RXLR effectors. For example, AlphaFold2 has been used to model *PiAvrblb2* (Figure 2C) and *PiAvr3a*, two well-known *P. infestans* effectors that show critical structural components implicated in immune evasion. Researchers can use the structural predictions of these effectors to find crucial interaction surfaces with R proteins, allowing them to create R proteins that can recognize a greater range of effector variations [164]. As more RXLR and NLR structures are predicted and validated, our understanding of the molecular basis of plant immunity will continue to grow.

More recently, AlphaFold2-Multimer (an adaptation of AF2 for protein complexes) has proven capable of predicting certain protein–protein interactions. Fick, Fick [166] applied AF2-Multimer to all known NLR–effector pairs across various plant pathosystems. Remarkably, AF2-Multimer was able to generate plausible models for complexes like RPP1–ATR1 and ROQ1–XopQ that align well with the cryo-EM structures (i.e., the predicted interface and overall orientation were correct). The study showed that AF2 could correctly distinguish true NLR–effector partners from non-partners by producing high-confidence interface models for true pairs, whereas random pairings gave low-confidence or unrealistic models. In addition, they also predicted binding affinity surrogates from these models, finding a narrow range of computed energies for true interactions. From the study, they suggested that this approach can identify novel NLR–effector interactions with ~99% accuracy when combined with machine learning classifiers.

The authors also concluded that despite the leaps made by AF2, caution is warranted. A confident AF2 prediction of a complex does not guarantee that a real interaction occurs in vivo; it could be picking up on subtle sequence correlations or biases in the training set. False positives can arise, especially for proteins with repeat domains like NLR LRRs, which could potentially align with an effector’s helical repeats to give a seemingly good interface. Conversely, some true interactions might be missed if the complex is very transient or if the binding causes large conformational changes that AF2, which tends to predict a single stable complex, cannot easily capture. For example, if an effector induces an allosteric change in an NLR rather than staying bound, AF2 might not model it accurately. Also, current AF2-Multimer is not guaranteed to get quaternary stoichiometry right; it might model a 1:1 complex when the real assembly is 4:4, or vice versa. Another limitation is that AF2 gives no direct information about kinetics or binding strength. It also does not model membrane interactions well; any effector or NLR that works at a membrane (like ZAR1 that forms pores or effectors that insert in membranes) is outside AF2’s main scope. Molecular dynamic (MD) simulations can complement static models by testing stability of the predicted complex and exploring conformational flexibility. Indeed, a possible future direction is to take an AF2-predicted Rpi–effector complex and run MDs to see if the interface remains stable and what interactions dominate; this could help refine the model or design mutations for experimental validation.

#### 4.3.4. Functional Genomics

Functional genomics methods, such as RNA sequencing (RNA-seq), transcriptome profiling, and dual RNA-seq, have shed light on the gene expression alterations caused by RXLR–NLR interactions. These tools enable researchers to investigate the transcriptional responses of both the pathogen and the host during infection, revealing the molecular pathways involved in immune responses and pathogen pathogenicity [167,168].

RNA sequencing (RNA-seq) has been applied to evaluate the transcriptomes of potato plants infected with *P. infestans*, revealing genes that are differentially expressed in response to infection [169]. This approach has shown that immune-related genes, particularly NLR genes, are activated, whereas susceptibility genes are suppressed [170]. RNA-seq has also been used to investigate *P. infestans* transcriptomes during infection, revealing the expression patterns of RXLR effectors and other virulence components [171].

Transcriptome profiling has revealed details about the temporal dynamics of RXLR–NLR interactions [172]. By studying gene expression variations at various phases of infection, researchers can discover critical regulatory networks and pathways involved in the immune response. This knowledge is critical for understanding the complicated interaction between the virus and host, as well as creating resistance-enhancing techniques.

#### 4.3.5. Dual RNA Sequencing Method

An accurate assessment of host and pathogen gene expression during infection is critical for understanding the molecular aspects of host–pathogen interactions. The transcriptome is a significant indicator of a cell’s physiological condition, whether healthy or infected [173]. Transcriptome analysis has long been a critical technique for understanding the molecular alterations that occur during pathogen infection in plants [174,175]. Such research has been limited to mRNA expression changes in either the pathogen or the infected host plant. However, the increased sensitivity of high-throughput RNA sequencing now allows for ‘dual RNA-seq’ studies, which capture all classes of coding and noncoding transcripts in both the pathogen and the host [176,177]. Dual RNA sequencing is a newly developed method for a comprehensive understanding of the host–pathogen interactions, involving simultaneous analysis of the gene expression changes in both the pathogen and host genomes [178].

The use of resistant cultivars of potato has been an effective way of controlling late blight disease; however, the host-driven selective pressure has resulted in the rapid mutation of RXLR genes, allowing the pathogen to escape host defense and cause infection [179]. As a result, understanding the expression profiles of RXLR and host genes via dual RNA-seq can be used as gateway for the molecular breeding of disease resistance. In understanding the RXLR gene–host interaction studies, the method has been employed to evaluate the compatible and incompatible interactions between *P. infestans* and potato cultivars, resistant and susceptible. In most cases, the study employs time-course dual RNA-seq. These studies unravel genes that are differentially expressed during the infection. Depending on the compatibility of the interactions, different genes can be upregulated or downregulated. Resistant potato activates a set of biotic stimulus responses and phenylpropanoid biosynthesis SEGs, including kirola-like protein, NLR, and disease resistance and kinase genes. Conversely, the susceptible potato upregulates more kinase, pathogenesis-related genes than resistant cultivars [180]. This method provides the number of RXLR effectors and NLR proteins that are expressed during infection. For instance, Li, Hu [173] reported that most expressed RXLR effector genes are suppressed during the first 24 h of infection but upregulated after 24 h. The study also showed that *P. infestans* induces more specifically expressed genes (SEGs), including RXLR effectors and cell wall-degrading enzyme (CWDE)-encoding genes in the compatible interaction. The study showed 559 RXLR effector proteins and 195 CRN effector genes with 208 and 49 being expressed during the infection process, respectively. Additionally, the study detected 91 NBS-LRR genes with varying expression profiles.

## 5. Genetic Improvements for Durable Resistance

From about 107 wild potato relatives, only 32 have been used to identify and map the *Rpi* genes. In addition, only a smaller number of these genes have been used in commercial potato cultivars. The main reason for this is that the introgression is limited to long breeding cycles and the high levels of heterozygosity across the potato genome. The wild relative that has been used most widely for the introgression of *R*-genes into potatoes is *S. demissum*, with a total of 11 *R* genes identified (*R1*, *R2*, *R3*, …, *R11*). Most of these R genes have been defeated by the pathogen, with the exception of *R8* and *R9a*, which are still deployed with other *Rpi* genes. An attempt to introgress the *Rpi* gene from *S. bulbocastanum* took more than 45 years with eventual success and registration occurring in 2006 and 2008 for two *P. infestans* resistant cultivars, Toluca and Bionica, respectively [25]. Other wild relatives of potato with a significant number of identified *Rpi* genes include *S. berthaultii* (*Rpi-ber1*, *Rpi-ber2*, *Rpi-ber1.2*, *Rpi-ber1.3*, *Rpi-ber1.4*), *S. bulbocastanum* (*Rpi-blb1*, *Rpi-blb2*, *Rpi-blb3*, *Rpi-blb4*), *S. stoloniferum* (*Rpi-sto1*, *Rpi-pta1*, *Rpi-sto2*, *Rpi-pta2*), *S. venturi* (*Rpi-vnt1.1*), and *S. americanum* (*Rpi-amr1*, *Rpi-amr3*, *Rpi-amr4*). A comprehensive review of these Rpi genes is given by Paluchowska, Sliwka [6] and Berindean, Taoutaou [130].

The breeding methods employed for the introgression of novel *Rpi* genes in commercial potato cultivars are faced with numerous challenges. This is due to the need for extensive back-crossing and rigorous selection pressures resulting in time-consuming and labor-intensive processes [181]. Different methods have been devised since then to circumvent the crossing barriers and speed up incorporating Rpi genes into commercial cultivars, including genetic engineering, hybrid breeding, and somatic hybridization. The approaches are extensively reviewed by Paluchowska, Sliwka [6].

Alternative approaches include the construction of synthetic *Rpi* genes, which involves domain shuffling, and prime and base DNA editing [182]. With recent breakthrough discoveries in the structure and molecular function of NLRs, there has been progress in the knowledge-guided molecular engineering of NLRs. The first studies were successful in extending or changing effector recognition specificities by structurally altering the NLR domains that directly bind to effectors. Monino-Lopez, Nijenhuis [80] in a study to determine the function of an allelic variant of *Rpi-chc.1.1*, that is, *Rpi-chc1.2* from *Solanum chacoense*, carried out domain swap and found out that the LRR domain harbors the recognition specificity of both *PiAvrchc1.1* and *PiAvrchc1.2*. The specificities reside in overlapping LRR subdomains and could not be combined into one active protein domain. From this study, Monino-Lopez, Nijenhuis [80] exchanged the LRR domain of Rpi-tub1.3_RH89-039-16, which does not recognize PexRD12/31 effectors, with the LRR domain of Rpi-chc1.2; the chimeric receptor was able to recognize both PexRD31-B and PexRD31-C. In addition to swapping the LRR domain between closely related NLR alleles, other approaches such as exchanging the polymorphic residues between the LRR domain have been performed to improve the recognition spectrum of the Rpi genes. For instance, in the same study by Monino-Lopez, Nijenhuis [80], it was discovered that the ability of Rpi-tub1.3_RH89-039-16 to recognize PexRD12/31 effectors (*PiAvrchc1.2*) after swapping the LRR domain was due to the exchange of the amino acid polymorphisms present in LRRs 16 to 19.

## 6. Conclusions

The molecular interplay between RXLR-encoded *PiAvr* genes and NLRs during *Phytophthora infestans* infection in potato represents a complex and dynamic interaction critical for plant immunity and disease development. The extensive diversity and evolution of RXLR effectors in *P. infestans* highlights the adaptability of pathogens and the challenges in developing durable resistance in potato crops.

On the other hand, the structural organization and evolutionary dynamics of NLR immune receptors underscore the importance of plant immune surveillance and the rapid evolution of defense mechanisms in response to pathogen pressure. The cloning and deployment of R genes for disease resistance offer promising strategies for mitigating the impact of late blight and other devastating plant diseases on potato cultivation.

Overall, a comprehensive understanding of the molecular mechanisms underlying the interaction between RXLR effectors and NLR immune receptors is essential for developing sustainable and effective strategies for disease management and crop improvement. Continued research in this field will contribute to the development of resilient potato varieties capable of withstanding the challenges posed by *Phytophthora infestans* and other pathogens, ensuring food security and agricultural sustainability worldwide.

## Figures and Tables

**Figure 1 ijms-26-08153-f001:**
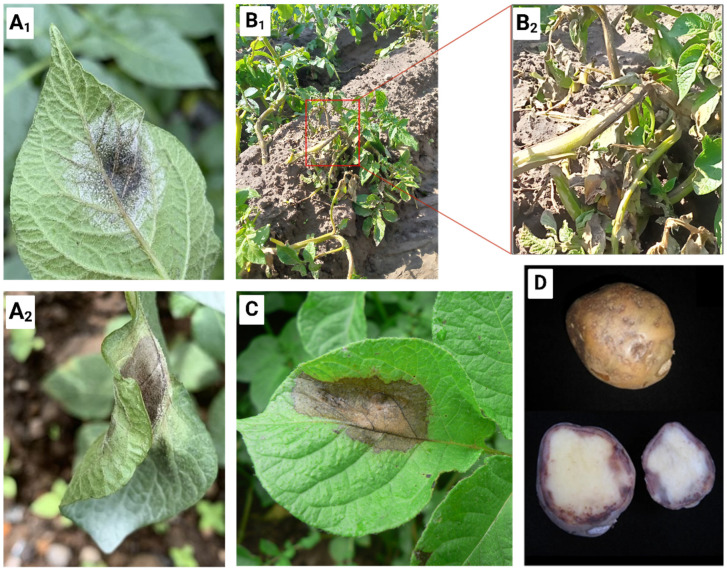
Typical symptoms of late blight disease (*Phytophthora infestans*). (**A_1_**,**A_2_**): White sporulation and lesion seen on the underside of the leaf. (**B_1_**,**B_2_**): Brown lesions on stem and petioles attacked by *P. infestans*. (**C**): An oily necrotic spot, surrounded by pale green on the upper side of the leaf. (**D**): Infected tuber with soft necrotic edges.

**Figure 2 ijms-26-08153-f002:**
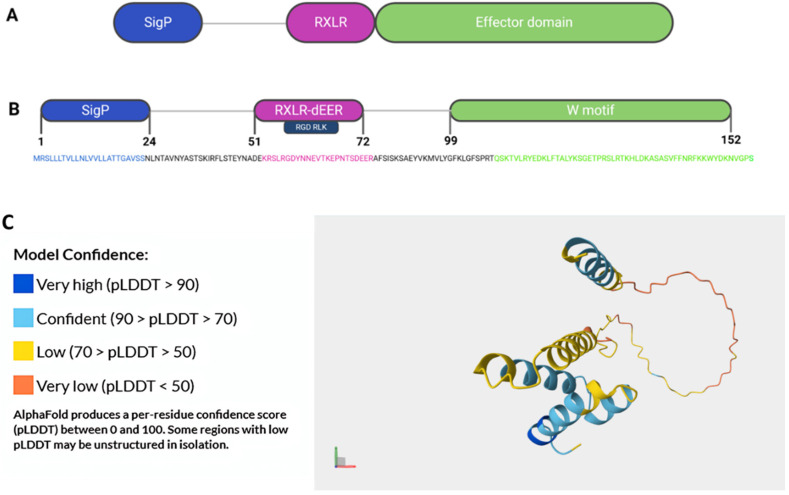
(**A**): Generalized domain structure of *P. infestans* RXLR effector proteins; (**B**): specific domain structure and amino acid sequence of the effector *PiAvrblb1*: the blue color region represents the signal peptide domain (1–24 aa), the purple region is the RxLR-dEER motif (51–72 aa) and RGD RLK-binding motif (54–56 aa), and green represents the W motif region (99–152 aa); (**C**): modular structure of *Piavrblb1*, as predicted by AlphaFold2 [44]. Created in BioRender.com.

**Figure 3 ijms-26-08153-f003:**
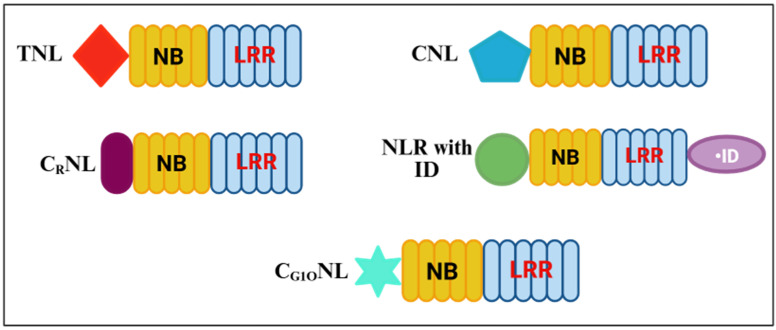
Schematic representation of the NLR protein domains. Different structures of NLR proteins identified in plants based on their N-terminal domain and additional integrated domain in the C-terminus; TNL (TIR-NLR), CNL (CC-NLR), C_R_NL (RPW8-NLR), and C_G1O_NL (G10-type CC-NLRs). Created in BioRender.com.

**Figure 4 ijms-26-08153-f004:**
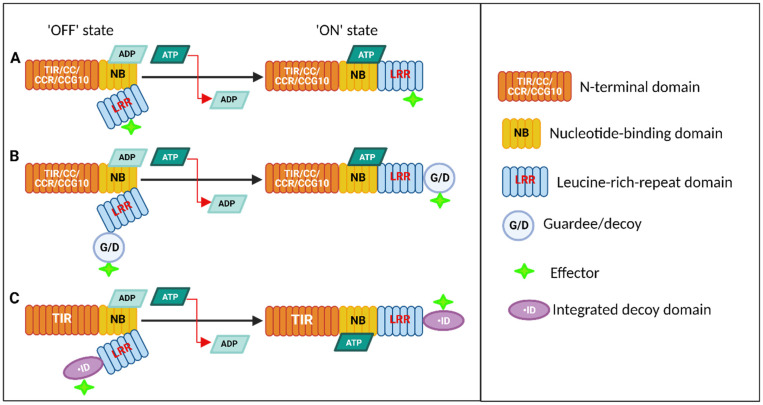
Schematic representation of tripartite structure of plant NLRs showing different types of pathogen recognition. The NLR consists of a diverse N-terminal domain, a central nucleotide-binding (NB) domain, and a C-terminal leucine-rich repeat (LRR) domain. NLRs are classified into four groups depending on the N-terminus: Toll-interleukin-1-receptor (TIR), coiled-coil (CC), and subsets of CC-NRL, resistance to powdery mildew 8 (CCR), and the G10-type CC-NLRs (CCG10). (**A**) NLRs recognize the pathogen directly through the LRR domain. (**B**) NLRs recognize pathogen effectors indirectly, a host guardee/decoy. (**C**) NLRs evolved with some acquiring unusual integrated decoy (ID) domains for pathogen recognition. Created in BioRender.com.

**Figure 5 ijms-26-08153-f005:**
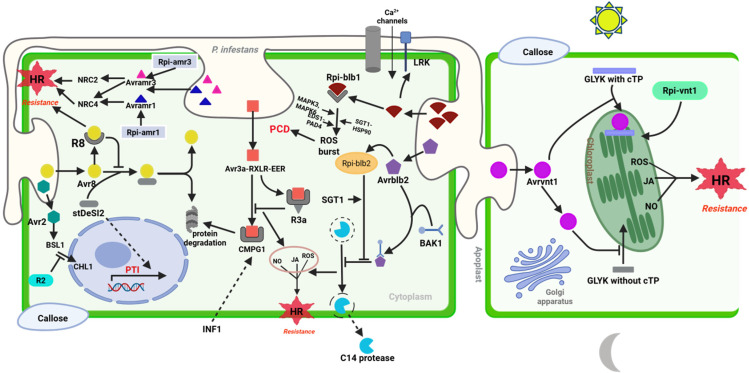
Schematic illustration of molecular interaction between selected Rpi genes and their cognate *PiAvr* genes. The figure illustrates the direct and indirect modes of effector recognition by NLRs, highlighting the specificity determinants involved in effector–NLR interactions and the subsequent activation of immune responses. Effector molecules from the pathogen (colored shapes) are recognized by plant resistance proteins (labeled *Rpi* genes), triggering defense mechanisms such as the hypersensitive response (HR; red stars), ROS production, and hormone signaling (JA and NO). Solid black arrows show activation; blunt ends show inhibition. Structures like the nucleus, chloroplast, and Golgi illustrate where immune responses are localized. Callose (white ovals) represents the physical reinforcement of cell walls. The blue DNA symbol represents transcriptional activation during pattern-triggered immunity (PTI). The continuous arrows denote activations; blunt lines indicate inhibition; and dashed line arrows indicated proposed links.

**Table 1 ijms-26-08153-t001:** Number and percentage of candidate R-genes belonging to different classes in *Solanum tuberosum* (based on [68]).

R Gene Class	*Solanum tuberosum*
Coiled coil (CC)	163 (0.466%)
Toll-interleukin-1 (TIR)	43 (0.123%)
Receptor-like protein (RLP)	403 (1.151%)
Receptor-like kinase (RLK)	301 (0.86%)
Others	421 (1.203%)

**Table 2 ijms-26-08153-t002:** Summary of some known cloned potato Rpi genes from wild *Solanum* relatives and their corresponding *Phytophthora infestans Avr* effectors.

Rpi Gene	Origin (*Solanum* Species)	Chr	Recognized Effector (Avr/RxLR)	Notes on Interaction	References
Wild tuber-bearing potato relatives					
*R1*	*S. demissum*	V	*Avr1*	*Avr1* triggers *R1*-mediated HR; deletion of C-terminal “T-region” enables HR. Race-specific resistance; first late blight *R* gene cloned. Overcome by modern virulent races.	Du, Weide [99]
*R2*		IV	*Avr2*	*Avr2* triggers *R2*-mediated HR, by binding host BSL1 and *R2* (guard model). Member of *R2/Rpi-blb3* locus on chr4; provides resistance to some *P. infestans* races but defeated by others.	Saunders, Breen [66], Rodewald and Trognitz [109]
*R3a*		XI	*Avr3a*	*Avr3a^KI^* variant triggers *R3a* HR; *Avr3a^EM^* escapes recognition.	Armstrong, Whisson [58]
*R3b* (*StR3b*)		XI	*Avr3b*	Distinct *R3* locus gene (82% identical to *R3a*) that specifically recognizes the *Avr3b* effector, which triggers HR; however, recognition requires SGT1/HSP90 (co-factor). *R3b*-mediated resistance is ineffective against races lacking *Avr3b*.	Li, Huang [110]
*R4*		XI	*Avr4*	C-terminal W-motifs of *Avr4* (W2+W1 or W3) are required for *R4* recognition.	Van Poppel, Jiang [111]
*R8*		IX	*Avr8*	*Avr8* destabilizes StDeSI2 through 26S proteosome inhibitor.	Vossen, van Arkel [98], Jiang, He [100]
*Rpi-blb1* (*RB*)	*S. bulbocastanum*	VIII	*IPI-O1/IPI-O2* (*Avr-blb1*)	Broad-spectrum resistance; recognizes class I/II IPI-O effectors (*Avrblb1*) that most isolates carry, by detecting when IPI-O1 binds the LecRK-I.9 (SEN1) receptor triggering HR. Homologous genes (e.g., *Rpi-sto1, Rpi-pta1*) in related species confer similar broad resistance.	Vleeshouwers, Rietman [39], Champouret, Bouwmeester [43]
*Rpi-blb2*		VI	*Avrblb2* (*IPI-O family*)	Confers strong broad resistance. *Avrblb2* binds to calmodulin at the plasma membrane. Triggers HR via *Rpi-blb2* which targets and inhibits host protease C14. *Rpi-blb2* is an Mi-1 homolog (shares ~82% identity) despite differing pathogen targets.	Oh, Young [112]
*Rpi-blb3*		IV	*Avr2*	Ortholog of *R2*; recognizes same *Avr2* effector and confers partial resistance (some isolates still could infect).	Vleeshouwers, Raffaele [113]
*Rpi-blb4*		V	*Unknown*	Newly identified resistance in *S. bulbocastanum* accession BLB7650, mapped to chromosome 5 via RenSeq. Confers late-blight resistance in that accession (details emerging).	Li, Kaur [114]
*Rpi-vnt1.1*	*S. venturii*	IX	*Avr_vnt1*	Confers broad resistance to most *P. infestans* isolates. *Avrvnt1* binds chloroplast enzyme GLYK, activating *Rpi-vnt1.1* in a light-dependent manner. Identical in sequence to *Rpi-phu1* from *S. phureja*. Ineffective only against a few rare *Avr-vnt1* virulent strains.	Foster, Park [77], Pais, Yoshida [115]
*Rpi-mcq1*	*S. mochiquense*	IX	*Avr2*	Unrelated CC-NLR; recognizes *Avr2* independently of *R2/Rpi-blb3* by employing BSL2 and BSL3. Identified via positional cloning in *S. mochiquense* and allele mining in related species. Represents an independent evolutionary origin of *Avr2* recognition on chr9.	Wang, Trusch [116]
*Rpi-pta1*	*S. papita*	VIII	*IPI-O1/IPI-O2* (*Avr-blb1*)	*Rpi-blb1* homolog from *S. papita*, which is functionally equivalent to *Rpi-sto1*. Provides broad resistance by recognizing *Avrblb1* (IPI-O) effectors.	Vleeshouwers, Rietman [39]
*Rpi-chc1.1*	*S. chacoense*	X	*Avr-chc1.1* (*PexRD12 family*)	One allele of the *Rpi-chc1* locus that recognizes multiple members of *PexRD12* (*Avr-chc1.1*) subfamily.	Monino-Lopez, Nijenhuis [80]
*Rpi-chc1.2*	*S. chacoense*	X	*Avr-chc1.2* (*PexRD31 family*)	Second allele at the *Rpi-chc1* locus; recognizes multiple members of *PexRD31* (*Avr-chc1.2*) subfamily. Shows no overlap with *Rpi-chc1.1* in effector recognition.	
*Rpi-phu1*	*S. phureja* (*diploid*)	IX	*Avr-vnt1*	Same gene as *Rpi-vnt1.1* (sequence-identical) mapping to chromosome 9. Discovered in Andean *S. phureja* landraces; confers broad resistance similarly to *Rpi-vnt1.1*.	Śliwka, Świątek [117]
*Rpi-sto1*	*S. stoloniferum*	VIII	*IPI-O1/IPI-O2* (*Avr-blb1*)	Functional homolog of *Rpi-blb1* and recognizes same IPI-O effectors, conferring broad resistance similar to *RB*. Cloned via effectoromics screens.	Vleeshouwers, Rietman [39]
Non-tuber-bearing Solanum species					
*Rpi-amr1*	*S. americanum*	XI	*Avramr1*	Resistance gene cloned from *S. americanum*. Mapped to chromosome 11; triggers immunity by recognizing the *AvrAmr1* effector (identified via long-read sequencing).	Witek, Lin [118]
*Rpi-amr3*		IV	*Avramr3 (conserved RXLR)*	Broad-spectrum resistance gene from *S. americanum*, effective against multiple *Phytophthora* species (or isolates). Mapped to chromosome 4 and cloned using SMRT RenSeq. Recognizes a conserved effector (*AvrAmr3*) present in several *Phytophthora spp*., conferring non-host-type resistance.	Lin, Olave-Achury [28]
*Rpi-amr4*		NA	*Avramr4*	Identified by sequencing, assembling high-quality genomes and defined the pan-NLRome of *S. americanum*. Recognizes cognate *P. infestan* RXLR effectors *PITG_22825* (*AVRamr4*).	Lin, Jia [84]

NA; chromosome location not identified.

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
