# Peer review of "Deciphering the Molecular Interplay Between RXLR-Encoded Avr Genes and NLRs During Phytophthora infestans Infection in Potato: A Comprehensive Review"

_ijms, 2025, doi:10.3390/ijms26178153_

Round 1
Reviewer 1 Report
Comments and Suggestions for Authors
The authors did a tremendous job in gathering the results of P infestans RXLR effectors and host NLR genes. However, it is hard to determine the topic of this review. Please do concentrate on some aspect of this problem and describe it. Currently, there is a huge amount of information about RXLR effectors and NLR proteins. If the authors would like to concentrate on the interactions between those two components, some additional information about the methods that can be used to study those interactions would be at hand. Also, a tabular summary of what is presented. Additionally, I would recommend adding some additional figures while describing the recognition mechanisms between effectors and resistance genes of the host. Some of the presented sections are disproportionately short in comparison to others. Please try to formulate a history of what the take-out message is from this manuscript. Below, I added some more detailed suggestions to the content of the manuscript, but I believe that the major point is the manuscript structure and what is its major topic is. Please try to address what a scientist who will read your manuscript might find most useful and concentrate on delivering that message. Unfortunately, in my opinion, the manuscript currently presents a huge amount of literature data which could be formulated into two separate review manuscripts, but currently lacks clarity of presentation; therefore, I recommend that this article undergo substantial revisions.
Line 35: Please briefly describe the pathogen biology and pathogenicity, referring to appropriate literature. P.i. is a hemibiotrophic pathogen, which adds to the complexity of the control of this pathogen. Additionally, it is important to mention the main routes of invasion since they greatly influence the plant response and control strategies.
Line 47: Provide some statistics for the losses
Line 52: Use one type of citation style,
Line 56: Please correct the figure caption
Line 93: unclear, please rewrite using short sentences.
Line 196: Please mark the motives from subfigure A on subfigure B
Line 291: Please summarize the presented examples in the table with protein names and molecular target
Line 345: Most potato cultivars have 48 chromosomes
Line 359: Please provide an example figure of the NLR protein
Line 381: NTP?
Line 383: Please verify this data you the p_loop domain binds ATP but is not responsible directly for it hydrolysis, as far as I know if there is some experimental evidence for this, please cite such studies
Line 599: all symbols and abbreviations used in the figure should be explained in the figure caption.
Line 821: While I do agree that alphaFold is a user-friendly tool for structural prediction, it is not the only predictive protein structure tool available. The authors should include a key drawback of the usage of such predictive tools, for example, for short proteins such as RXLR effectors, as shown in Figure 1, which
Line 825: It is computational methods or predictive methods, although it is true that AlphaFold popularized predictive methods, also while this is a very comprehensive review presenting multiple aspects, it fails to mention any experimental method for confirming protein interaction.
For figures created in BioRender.com, please provide a copyright permission
Reviewer 2 Report
Comments and Suggestions for Authors
This comprehensive review elucidates the intricate molecular interplay between Phytophthora infestans RXLR effectors and potato NLR immune receptors, highlighting their structural diversity, evolutionary dynamics, and recognition mechanisms. The interdisciplinary perspective bridges fundamental research with applied breeding strategies, offering a roadmap for sustainable disease management in potato cultivation. The content of this manuscript is detailed and well-organized.
There are some minor comments:
- Lines 15-18: “This review provides a comprehensive analysis of the structural characteristics, functional diversity, evolutionary dynamics of RXLR effectors, and the mechanisms by which NLR receptors recognize and respond to them” is better.
- Lines 12-15: “The battle...is driven by molecular interplay of RXLR-encoded PiAvr effectors with NLR immune receptors.” is better.
- Line 36: "Potato (S. tuberosum) is a crucial global staple crop." is better
- Lines 44, 45: “Phytophthora infestans”. The full name should be used for the first time, and "P. infestans" can be used in the following text.
- Lines 125-127: "Pathogens evolved sophisticated arsenals to manipulate host cells during plant-pathogen coevolution." is better.
- Lines 127-130: "Oomycete pathogens employ secreted RXLR effectors, named for their conserved N-terminal Arg-X-Leu-Arg (RXLR) motif." is better.
- Line 146: Delete “plant’s”
- Line 334 Delete “inhibiting the activation of immunological responses,”
- Line 320: Delete “from P. infestans”. Modify “translocate” to “translocates”
Author Response
We would like to thank Reviewer 2 for all his/her comments and feedback. All the improvements suggested were included in the manuscript.
Round 2
Reviewer 1 Report
Comments and Suggestions for Authors
I want to thank the authors for addressing my comments. However, to provide a quick reevaluation of your manuscript, I would need a version with tracked changes; otherwise, I would require more time to evaluate it. The citation style still does not match the IJMS guidelines, and there is no tabular summary present in the submitted version of the manuscript.
Author Response
Comment 1: I want to thank the authors for addressing my comments. However, to provide a quick reevaluation of your manuscript, I would need a version with tracked changes; otherwise, I would require more time to evaluate it. The citation style still does not match the IJMS guidelines, and there is no tabular summary present in the submitted version of the manuscript.
Response 1: Thank you for the comment. We have provided version with tracked changes in addition to the clean version. We have also updated the citation style to match the IJMS guidelines. Finally, we have also incorporated the tabular summary in the main manuscript.
Round 3
Reviewer 1 Report
Comments and Suggestions for Authors
I want to thank the authors for revising their manuscript and positively responding to most of my comments. An additional figure facilitates the interpretation of this manuscript. I would still recommend adding more summarizing schemes and tables to this long and insightful publication. Please correct the remaining citations according to the MDPI citation style. Concluding I find this manuscript extremely informative with a lot of interesting technical information; however, in my opinion, summarizing schemes and tables would increase its clarity. The content of this manuscript is much easier to follow after implementing corrections. My only remaining concern is the absence of the summarizing Table 2 in the main text. In my opinion, this is a key element for this manuscript, which helps to follow its content. Therefore, my recommendation for this manuscript is to undergo minor revisions to correct the citation formatting and consider uploading Table 2 to the main text.
Line 787: The pathogen
Author Response
Comment 1: I want to thank the authors for revising their manuscript and positively responding to most of my comments. An additional figure facilitates the interpretation of this manuscript. I would still recommend adding more summarizing schemes and tables to this long and insightful publication. Please correct the remaining citations according to the MDPI citation style. Concluding I find this manuscript extremely informative with a lot of interesting technical information; however, in my opinion, summarizing schemes and tables would increase its clarity. The content of this manuscript is much easier to follow after implementing corrections. My only remaining concern is the absence of the summarising Table 2 in the main text. In my opinion, this is a key element for this manuscript, which helps to follow its content. Therefore, my recommendation for this manuscript is to undergo minor revisions to correct the citation formatting and consider uploading Table 2 to the main text.
Response 1: Thank you for pointing out this insightful point. We agree with this comment, and we have incorporated a summarising table 2 in the main document. Additionally, we have corrected all the citations which were not in accordance with the MDPI citation style.